# Development Progress of 3–5 μm Mid-Infrared Lasers: OPO, Solid-State and Fiber Laser



Tingwei Ren [1], Chunting Wu [1,*], Yongji Yu [1], Tongyu Dai [2], Fei Chen [3] and Qikun Pan [3]

1   Jilin Key Laboratory of Solid-State Laser Technology and Application, School of Science, Changchun University of Science and Technology, Changchun 130022, China; 2020100071@mails.cust.edu.cn (T.R.); yuyongjiyuyongji@163.com (Y.Y.)
2   National Key Laboratory of Tunable Laser Technology, Harbin Institute of Technology, Harbin 150001, China; daitongyu@hit.edu.cn
3   State Key Laboratory of Laser Interaction with Matter, Changchun Institute of Optics, Fine Mechanics and Physics, Chinese Academy of Sciences, Changchun 130033, China; feichenny@126.com (F.C.); panqikun2005@163.com (Q.P.)
*   Correspondence: bigsnow1@126.com

**Abstract:** A 3–5 μm mid-infrared band is a good window for atmospheric transmission. It has the advantages of high contrast and strong penetration under high humidity conditions. Therefore, it has important applications in the fields of laser medicine, laser radar, environmental monitoring, remote sensing, molecular spectroscopy, industrial processing, space communication and photoelectric confrontation. In this paper, the application background of mid-infrared laser is summarized. The ways to realize mid-infrared laser output are described by optical parametric oscillation, mid-infrared solid-state laser doped with different active ions and fiber laser doped with different rare earth ions. The advantages and disadvantages of various mid-infrared lasers are briefly described. The technical approaches, schemes and research status of mid-infrared lasers are introduced.

**Keywords:** mid-infrared; optical parametric oscillator (OPO); solid-state lasers; fiber lasers

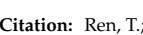

## 1. Introduction

Laser has been an important invention in the history of human science since the 20th century, following atomic energy, semiconductors and computers, known as "the fastest knife", "the most accurate ruler" and "the brightest light". Laser has been widely used and recognized in production and science because of its incomparable advantages over ordinary light sources. After 60 years of research and development, laser-related technologies, products and services have spread all over the world, forming a rich and huge laser industry. It is widely used in material processing, communication, optical storage, medical and beauty technologies, research and military developments, instruments and sensors, entertainment display, additive manufacturing and other areas of the national economy. In particular, high-performance 3–5 μm mid-infrared laser in the atmospheric window has important application value and prospect in laser imaging, chemical remote sensing, the medical field, environmental protection and civil and military fields [1].

At present, the technical ways to realize the mid-infrared laser output at a 3–5 μm band mainly include indirect conversion and direct generation. The indirect conversion is mainly based on the nonlinear frequency conversion crystal to generate mid-infrared laser by using an optical parametric oscillator, and the direct generation of stimulated radiation mainly includes quantum lasers, chemical lasers, gas lasers, solid-state lasers and fiber lasers [2]. The characteristic analysis of various ways to realize mid-infrared laser output is shown in Table 1.

**Table 1.** Comparative analysis of research approaches for realizing mid-Infrared 3–5 μm band.

| Method | Technology | Classification | Advantage | Disadvantage |
|---|---|---|---|---|
| Indirect conversion | Optical Parametric Oscillator | $LiNbO_3$, PPLN, MgO: PPLN, KTP, KTA, $ZnGeP_2$, $AgGaSe_2$, $AgGaS_2$ | high energy and efficiency and excellent spectral characteristics | system stability and beam quality should be improved |
| Directly produced | Quantum Cascade Laser | InAs, AlSb | wider transmission bandwidth | low output power and poor beam quality |
| | Chemical laser | HF, COIL | good beam quality | high prices, toxic products |
| | Gas laser | CO, $CO_2$ | high power and long service life | high temperature explosive, large volume and high cost |
| | solid-state laser | Fe: ZnSe, Cr: ZnSe | absorption bandwidth, wide tuning range and good beam quality | limited by temperature |
| | Fiber Laser | $Er^{3+}$: ZBLAN, $Ho^{3+}$: ZBLAN, $Dy^{3+}$: ZBLAN | small transmission loss and stable property | narrow tuning range |

As shown in the table, in view of the characteristics of the simple structure, small size, easy application and so on, this paper focuses on the introduction on the research of an optical parametric oscillator, excessive metal doped solid-state lasers and a fiber laser whose gain medium is soft glass.

## 2. Mid-Infrared Optical Parametric Oscillation Laser (OPO)

The optical parametric oscillation laser (OPO) is one of the main ways to realize a mid-infrared laser output of 3–5 μm and is composed of nonlinear crystal, a pump source and a resonant cavity, as shown in Figure 1. It can reach an output band that cannot be realized by traditional lasers and has many advantages, such as a wide tuning range, simple structure, high output power, narrow linewidth, etc. [3]. With the emergence of various nonlinear crystals, the optical parametric oscillator has achieved important breakthroughs and opened new application prospects, which has once again become a research hot spot of scholars in the world. According to the different nonlinear crystal materials, the mid-infrared laser based on optical parametric oscillation is classified as follows.

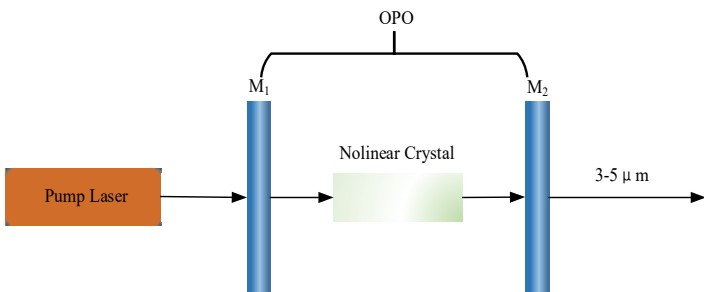

**Figure 1.** Schematic diagram of optical parametric oscillator.

### 2.1. LiNbO₃, PPLN, MgO-Doped PPLN Optical Parametric Oscillator

The optical parametric oscillators of lithium niobate crystals can be divided into pure lithium niobate ($LiNiO_3$), periodically poled lithium niobate (PPLN) and periodically po-larized lithium niobate doped with MgO (MgO-doped PPLN) optical parametric oscillators based on different crystals. The specific evolution process is shown in Figure 2. In order to improve the damage threshold and stability of the crystal, PPLN is used instead of the traditional $LiNiO_3$ crystal. While in order to further improve the damage threshold of the PPLN crystal, MgO-doped PPLN crystal was born.

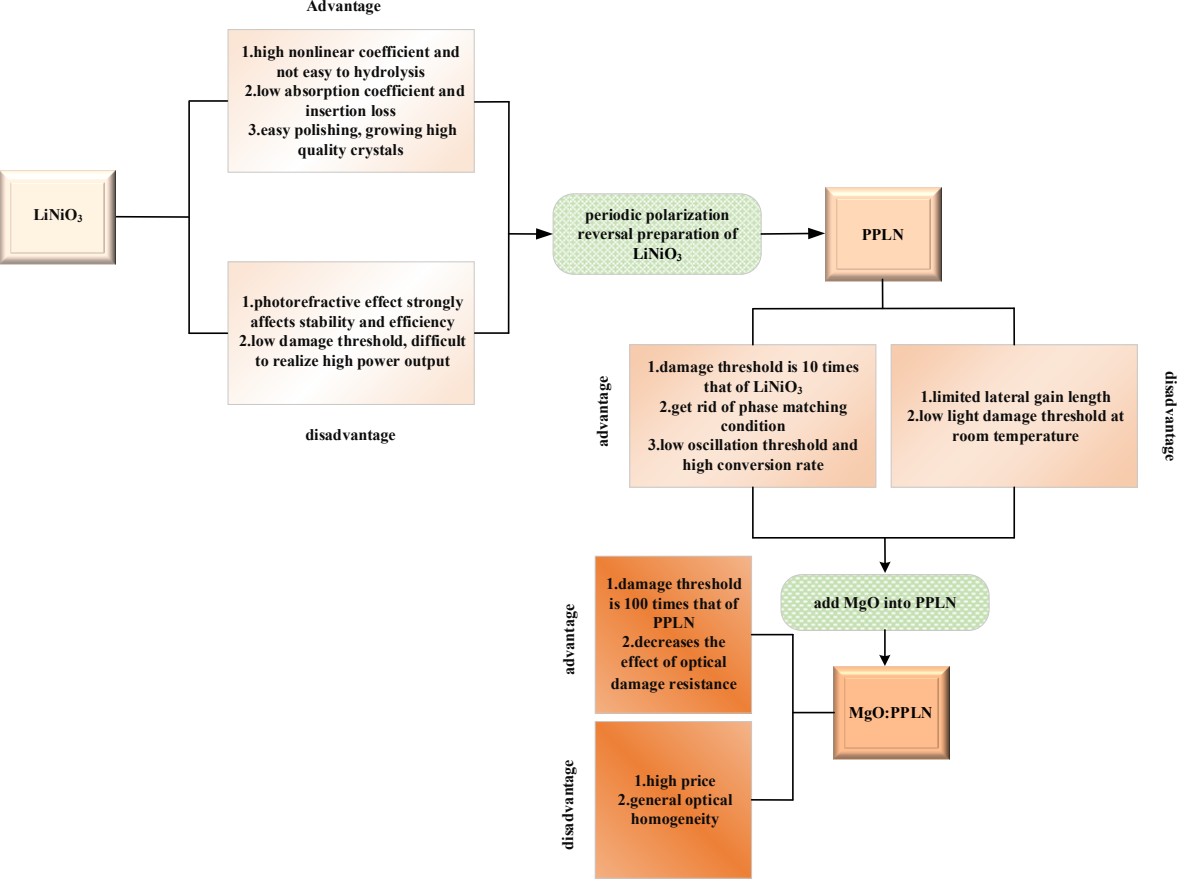

**Figure 2.** Evolution process of LiNiO$_3$ crystal.

From Figure 2, we can see that LiNbO$_3$, PPLN and MgO-doped PPLN all have their own advantages and disadvantages. The technology of periodically polarized crystals has been gradually developed and perfected with the increasing research of scholars. The research statuses of LiNiO$_3$, PPLN and MgO-doped PPLN optical parametric oscillation lasers are shown in Table 2.

**Table 2.** Research and development status.

| Crystal | Year | Research Establishment | Crystal Parameter | Pump Source | Mid-Infrared Output Parameter | Reference |
|---|---|---|---|---|---|---|
| LiNbO$_3$ | 2000 | North China Institute of Optoelectronic Technology | $10 \times 10 \times 30$ mm$^3$ | 1.06 μm Nd: YAG | Output wavelength 3.76 μm Repetition rate 5 Hz Average power 35 mW Optical efficiency 6% | [4] |
| | 2003 | Harbin Institute of Technology | No mention | 1.06 μm Nd: YAG | Output wavelength 3.41 μm Repetition rate 10 Hz Average power 12 mW Optical efficiency 4.5% | [5] |
| | 2006 | Sichuan University | $13 \times 13 \times 50$ mm$^3$ | 1.064 μm Nd: YAG | Output wavelength 3.06 μm Repetition rate 1 Hz Average power 15 mW Optical efficiency 10% | [6] |
| PPLN | 2011 | Photonics Center | $10 \times 20 \times 0.5$ mm$^3$ | 1.064 μm Nd: YVO$_4$ | Output wavelength 4.5 μm Average power 1.1 W Optical efficiency 7.5% | [7] |
| | 2012 | Tianjin University | $24 \times 8 \times 1$ mm$^3$ | 1.064 μm Nd: YVO$_4$ | Output wavelength 3.66 μm Average power 1.54 W Optical efficiency 7% | [8] |

**Table 2.** *Cont.*

| Crystal | Year | Research Establishment | Crystal Parameter | Pump Source | Mid-Infrared Output Parameter | Reference |
|---|---|---|---|---|---|---|
| | 2015 | Huazhong Institute of Optoelectronics Technology | $40 \times 10 \times 1$ mm$^3$ | 1.064 μm Nd: GdVO$_4$ | Output wavelength 3.81 μm Repetition rate 10 kHz Average power 5.4 W Optical efficiency 15.88% | [9] |
| | 2019 | Barcelona Institute of Science and Technology | 42 mm length 1 thick | 1.064 μm Yb$^{3+}$ fiber | Output wavelength 3.340 μm Average power 3.5 W Optical efficiency 9.5% | [10] |
| MgO -doped PPLN | 2008 | Harbin Institute of Technology | $50 \times 8.2 \times 1$ mm$^3$ 5% mol | 1.047 μm Nd: YAG | Output wavelength 3.26 μm Repetition rate 10 kHz Average power 0.46 W Optical efficiency 15.3% | [11] |
| | 2008 | China Academy of Engineering Physics | No mention 5% mol | 1.064 μm Yb$^{3+}$ fiber | Output wavelength 3.7 μm Average power 3.2 W Optical efficiency 18% | [12] |
| | 2010 | Tsinghua University | $5 \times 1 \times 30$ mm$^3$ No mention | 1.064 μm Nd: YVO$_4$ | Output wavelength 3.164 μm Repetition rate 76.8 kHz Average power 4.3 W Optical efficiency 17.1% | [13] |
| | 2012 | University of Southampton | $50 \times 2 \times 2$ mm$^3$ No mention | 1.064 μm Yb$^{3+}$ fiber | Output wavelength 3.82 μm Repetition rate 100 kHz Average power 5.5 W Optical efficiency 45% | [14] |
| | 2014 | Changchun University of Science and Technology | $50 \times 2 \times 2$ mm$^3$ 5% mol | 1.064 μm Nd: GdVO$_4$ | Output wavelength 3.85 μm Repetition rate 200 kHz Average power 1.82 W Optical efficiency 21.3% | [15] |
| | 2014 | Zhejiang University | $50 \times 1 \times 10$ mm$^3$ 5% mol | 1.064 μm Yb$^{3+}$ fiber | Output wavelength 3.99 μm Average power 2.1 W Optical efficiency 5.2% | [16] |
| | 2016 | Université Paris-Saclay | 1 length 5% mol | 1.55 μm Yb$^{3+}$ fiber | Output wavelength 3.07 μm Repetition rate 125 kHz Average power 1.25 W Optical efficiency 17.9% | [17] |
| | 2017 | Imperial College London | $40 \times 10 \times 1$ mm$^3$ 5% mol | 1.065 μm Yb$^{3+}$ fiber | Output wavelength 3.35 μm Repetition rate 1 MHz Average power 6.2 W Optical efficiency 24.3% | [18] |
| | 2018 | Changchun University of Science and Technology | $1 \times 8.6 \times 50$ mm$^3$ 5% mol | 1.06 μm Nd: YVO$_4$ | Output wavelength 3.2 μm Average power 1.72 W Optical efficiency 7.17% Output wavelength 3.5 μm Average power 1.39 W Optical efficiency 5.4% Output wavelength 3.8 μm Average power 1.39 W Optical efficiency 3.1% Output wavelength 4.1 μm Average power 0.72 W Optical efficiency 1.84% | [19] |
| | 2020 | Xinjiang Normal University | $40 \times 10 \times 2$ mm$^3$ 5% mol | 1.064 μm Nd: YAG | Output wavelength 3.4 μm Repetition rate 50 Hz Average power 1.075 W Optical efficiency 10.2% | [20] |
| | 2020 | Shandong University | $25 \times 3 \times 1$ mm$^3$ 5% mol | 1.937 μm Tm: YAP | Output wavelength 3.87 μm Repetition rate 6 kHz Average power 1.2 W Optical efficiency 19.4% | [21] |

**Table 2.** *Cont.*

| Crystal | Year | Research Establishment | Crystal Parameter | Pump Source | Mid-Infrared Output Parameter | Reference |
|---------|------|------------------------|-------------------|-------------|-------------------------------|-----------|
| | 2021 | Changchun University of Science and Technology | $30 \times 2 \times 5$ mm$^3$ 5% mol | 1.064 μm Yb$^{3+}$ fiber | Output wavelength 3.8225 μm Repetition rate 1 MHz Average power 2.06 W Optical efficiency 11.38% | [22] |
| | 2021 | Shandong University | $10 \times 1 \times 50$ mm$^3$ 5% mol | 1.064 μm Yb$^{3+}$ fiber | Output wavelength 3.4 μm Repetition rate 5 kHz Average power 3.68 W Optical efficiency 37% | [23] |

It can be seen from the table that the output power, wavelength and conversion efficiencies of periodically poled crystals have been improved substantially from LiNbO$_3$ to MgO-doped PPLN.

### 2.2. KTiOPO$_4$ and KTiOAsO$_4$ Optical Parametric Oscillator

KTP crystal and KTA crystal belong to the isologue, the symmetrical structure of the 2 m point group, which has high hardness and excellent optical properties. They are nonlinear optical materials widely used in frequency conversions. The descriptions of the two crystals are shown in Figure 3.

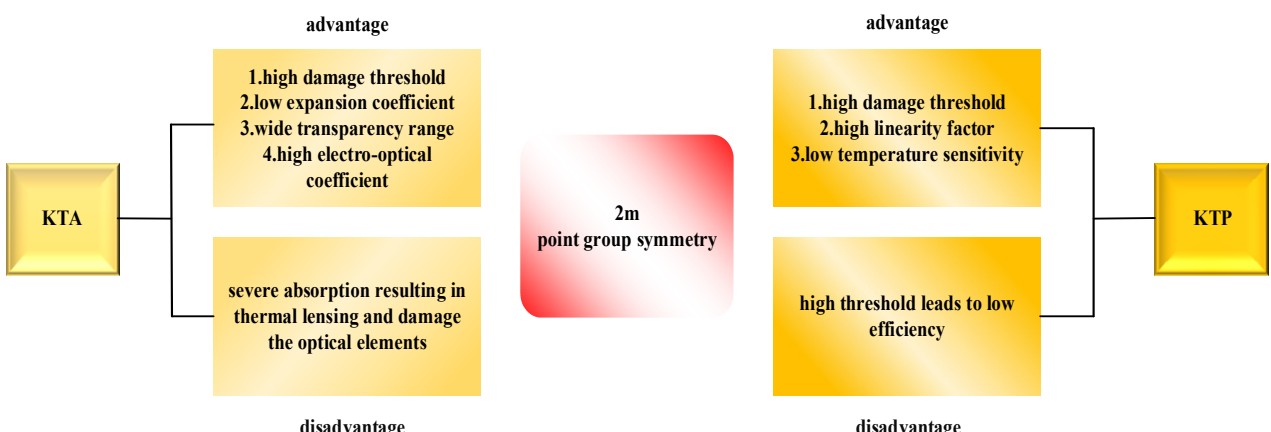

**Figure 3.** Description diagram of KTP and KTA crystals.

It can be seen from the Figure 3 that both KTP and KTA have the characteristics of a high damage threshold. However, compared with KTP crystal, the physicochemical property of KTA crystal is more stable and overcomes the absorption band of KTP crystal, which is near 3.4 μm. Both crystals have made prominent contributions to the high repetition frequency and high-energy mid-infrared output, and the excellent characteristics of KTP and KTA crystals determine the wide range of their applications. The research progress of KTP and KTA crystals in the mid-infrared band is shown in Table 3.

Numerous institutions for KTP and KTA crystals research have been reported. They have a wide variety of pump sources, and the operation modes are various. According to the latest research, they have achieved high-power and high-quality laser output.

**Table 3.** Research and development status.

| Crystal | Year | Research Establishment | Crystal Parameter | Pump Source | Mid-Infrared Output Parameter | References |
|---|---|---|---|---|---|---|
| KTP | 2003 | Harbin Institute of Technology | $7 \times 7 \times 25$ mm$^3$ | 1.06 μm Nd: YAG | Output wavelength 3.29 μm Average power 2 mW Repetition rate 1 Hz Optical efficiency 5% | [5] |
| | 2016 | The Czech Academy of Sciences | 16.5 mm length 1 mm thickness | 0.976 μm Yb$^{3+}$ fiber | Output wavelength 3.225 μm Repetition rate 100 MHz | [24] |
| | 2018 | Humboldt-Universität zu Berlin | 2 mm thickness | 1.028 μm Yb: KGd(WO$_4$)$_2$ | Output wavelength 3.13 μm Average power 780 mW Repetition rate 100 kHz Optical efficiency 12% | [25] |
| | 2021 | Chinese Academy of Sciences | $2 \times 4 \times 4$ mm$^3$ | 1.03 μm Yb:KGW | Output wavelength 3.17 μm Average power 1.03 W Repetition rate 15 MHz Optical efficiency 14.7% | [26] |
| KTA | 2010 | Chinese Academy of Sciences | $5 \times 5 \times 25$ mm$^3$ | 1.064 μm Nd: YAG | Output wavelength 3.467 μm Average power 84 mW Repetition rate 100 Hz Optical efficiency 14% | [27] |
| | 2011 | Norla Institute of Technical Physics | $7 \times 7 \times 20$ mm$^3$ | 1.064 μm Nd: YAG | Output wavelength 3.475 μm Average power 2.125 W Repetition rate 25 Hz Optical efficiency 14.3% | [28] |
| | 2013 | Whenzhou University | $5 \times 5 \times 20$ mm$^3$ | 0.808 μm Nd: YLF | Output wavelength 3.440 μm Average power 0.335 W Optical efficiency 5.6% | [29] |
| | 2013 | Tsinghua University | $10 \times 10 \times 20$ mm$^3$ | 1.06 μm Nd: YAG | Output wavelength 3.75 μm Average power 600 mW Repetition rate 10 Hz Optical efficiency 7.54% | [30] |
| | 2016 | Shanghai Institute of Optics and Fine Mechanics, the Chinese Academy of Sciences | $3 \times 2.5 \times 2$ mm$^3$ | 0.8 μm Ti: sapphire | Output wavelength 3.27 μm Average power 82 mW Repetition rate 1 kHz Optical efficiency 14.6% | [31] |
| | 2018 | Chinese Academy of Sciences | 2 mm length | 1.03 μm Yb: KGW | Output wavelength 3.05 μm Average power 1.31 W Repetition rate 151 MHz Optical efficiency 18.7% | [32] |
| | 2020 | U.S. Army Combat Capabilities Development Command | $6 \times 6 \times 20$ mm$^3$ | 1.06 μm Nd: YAG | Output wavelength 3.5 μm Average power 0.242 W Repetition rate 20 Hz Optical efficiency 35.5% | [33] |
| | 2021 | Shandong University | $10 \times 10 \times 33$ mm$^3$ | 1.064 μm Nd: YAG | Output wavelength 3.47 μm Average power 6.4 W Repetition rate 100 Hz Optical efficiency 43.6% | [34] |

## 2.3. AgGaSe$_2$ and AgGaS$_2$ Optical Parametric Oscillator

AgGaSe$_2$ and AgGaS$_2$ are semiconductor chalcopyrite symmetry crystals. Both crystals are transparent in infrared, and they have been used for a long time in the mid-infrared band. The descriptions of two crystals are shown in Figure 4.

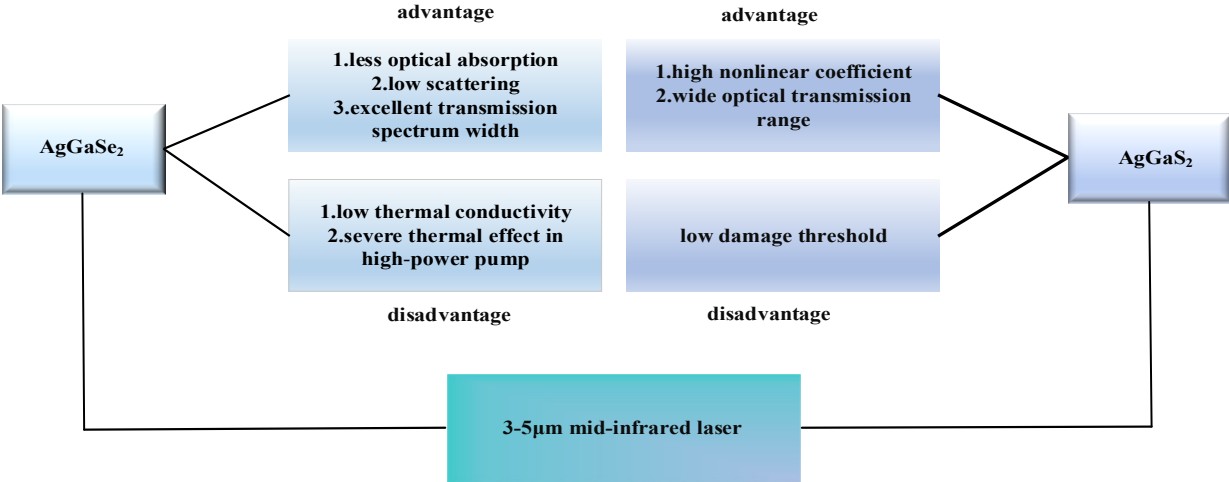

**Figure 4.** Crystal description diagram.

For AgGaSe$_2$ and AgGaS$_2$ crystals, the biggest defect is that the damage interpretation value is generally low, which cannot meet the needs of high repetition rates and maximum energy output.

In the early stage, the research on AgGaSe$_2$ and AgGaS$_2$ crystals was also extensive; the research and development status are shown in Table 4.

**Table 4.** Research and development status.

| Crystal | Year | Research Establishment | Crystal Parameter | Pump Source | Mid-Infrared Output Parameter | References |
|---|---|---|---|---|---|---|
| AgGaSe$_2$ | 2000 | The University of Burdwan | 9 mm thickness | 2 µm CO$^2$ laser | Output wavelength 3.5 µm<br>Average power 6 mW<br>Repetition rate 1 Hz<br>Optical efficiency 2.4% | [35] |
| | 2009 | Changchun Institute of Optics, Fine Mechanics and Physics | 18 × 18 × 52 mm$^3$ | 9.3 µm TEACO$^2$ laser | Output wavelength 4.65 µm<br>Average power 3.9 W<br>Repetition rate 100 Hz<br>Optical efficiency 56% | [36] |
| | 2013 | Huazhong University of Science and Technology | 5 × 5 × 13 mm$^3$ | 9.6 µm CO$^2$ laser | Output wavelength 3.2 µm<br>Average power 4 kW<br>Repetition rate 1 Hz<br>Optical efficiency 0.14% | [37] |
| AgGaS$_2$ | 1984 | Stanford University | 2 × 1 × 0.5 mm$^3$ | 1.064 µm Nd: yttrium | Output wavelength 4 µm<br>Average power 5 mW<br>Repetition rate 10 Hz<br>Optical efficiency 16% | [38] |
| | 1997 | DSO National Laboratories | 2 × 0.7 × 0.7 mm$^3$ | 1.064 µm Nd: YAG | Output wavelength 4.2 µm<br>Repetition rate 10 Hz<br>Optical efficiency 10% | [39] |
| | 1999 | American Institute of Physics | 20 × 7 × 10 mm$^3$ | 1.06 µm Nd: YAG | Output wavelength 3.9 µm<br>Average power 4 mW<br>Repetition rate 10 Hz<br>Optical efficiency 22% | [40] |
| | 2006 | Jilin University | 10 × 7 × 20 mm$^3$ | 1.06 µm Nd: YAG | Output wavelength 4 µm<br>Average power 12 mW<br>Repetition rate 20 Hz<br>Optical efficiency 3.5% | [41] |

It can be seen from the existing reports that the output efficiency based on these two crystals to realize mid-infrared laser is low, and the maximum energy that can be obtained

is also relatively small. This may be the reason why there are almost no literature reports about realizing mid-infrared laser output based on these two nonlinear crystals in the pas decade.

## 2.4. ZnGeP$_2$ Optical Parametric Oscillator

ZnGeP$_2$ crystal is the most important nonlinear crystal in optical parametric oscillator technology. The description of it is shown in Figure 5.

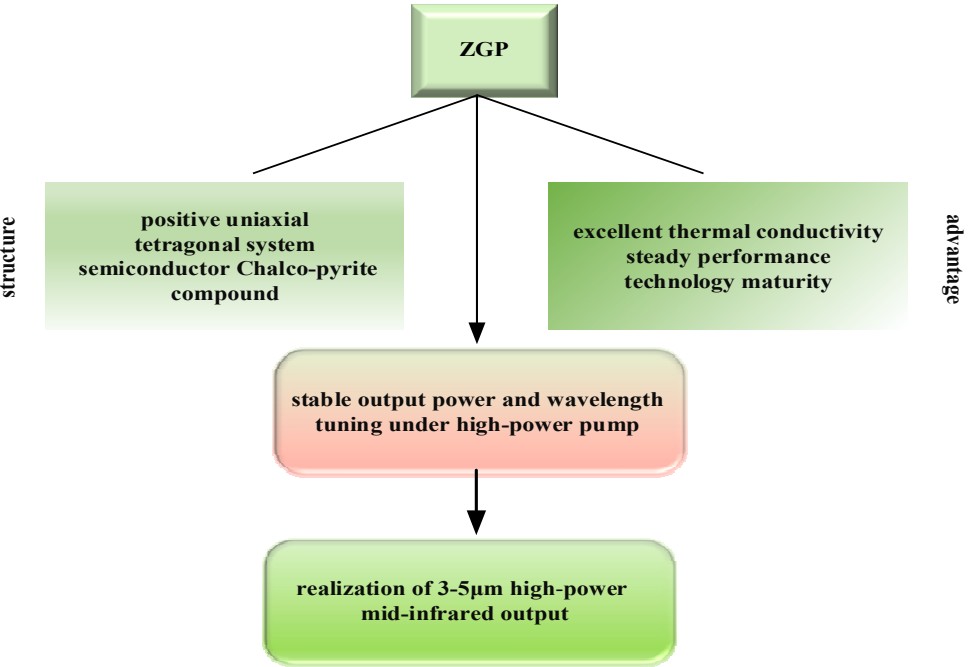

**Figure 5.** Description diagram of ZGP crystal.

For the ZnGeP$_2$ crystal, its good physical and chemical properties, high thermal conductivity and damage threshold have achieved its advantages when operating in a high-power environment. Therefore, it is the best nonlinear crystal for a high-power, 3~5 μm mid-infrared OPO.

The ZnGeP$_2$ crystal has been deeply studied by many scholars because of its excellent characteristics. The research development is shown in Table 5.

According to the literature, the best results of mid-infrared laser output based on ZGP crystal are an average output power of 103 W at a frequency of 10 kHz. The optical efficiencies are 78% and 44.2% with an output wavelength of 4.6 μm and 4.57 μm, respectively.

As mentioned above, several optical parametric oscillators for mid-infrared (3–5 μm) output are discussed. The properties parameters of mid-infrared nonlinear optical crystals are shown in Table 6.

The nonlinear crystals mentioned above have transmittance in the mid-infrared of 3–5 μm, which are currently widely studied in the world. Compared with LiNiO$_3$ and PPLN, MgO-doped PPLN crystal owns a larger damage threshold, and now it has become a research hotspot. However, the thermal conductivity of KTP, AgGaSe$_2$ and AgGaS$_2$ are relatively small, which will induce serious thermal effect under high-power operation and even cause the damage of crystals. Therefore, the output and applications of high-power mid-infrared in the future are limited. The thermal conductivity is smallest, and the damage threshold is the highest of ZGP crystal, which may be the reason why the output power is largest among these nonlinear crystals. It has a compact laser structure, the advantages of a wide tuning range of output wavelength and so on. Therefore, it can be said that the realization of mid-infrared laser output based on ZGP crystal is mainstream through an indirect way.

**Table 5.** Research and development status.

| Year | Research Establishment | ZGP Crystal Parameter | Pump Source | Mid-Infrared Output Parameter | References |
|---|---|---|---|---|---|
| 2010 | Norwegian Defence Research Establishment | $8.5 \times 6 \times 8$ mm$^3$ | 2.1 μm Ho: YAG | Output wavelength 4.5 μm<br>Average power 22 W<br>Repetition rate 45 kHz<br>Optical efficiency 58% | [42] |
| 2011 | China Academy of Engineering Physics | $8 \times 6 \times 18$ mm$^3$ | 2.1 μm KTP OPO | Output wavelength 4.32 μm<br>Average power 5.7 W<br>Repetition rate 8 kHz<br>Optical efficiency 46.6% | [43] |
| 2013 | Australian National University | No mention | 2.09 μm Ho: YAG | Output wavelength 3.5 μm<br>Average power 10.6 W<br>Repetition rate 35 kHz<br>Optical efficiency 69% | [44] |
| 2014 | University of Central Florida | $5 \times 4 \times 12$ mm$^3$ | 1.98 μm Tm: fiber | Output wavelength 3.7 μm<br>Average power 2.8 W<br>Repetition rate 4 kHz<br>Optical efficiency 8% | [45] |
| 2014 | Harbin Institute of Technology | $6 \times 6 \times 23$ mm$^3$ | 2.1 μm Ho: YAG | Output wavelength 4.5 μm<br>Average power 41.2 W<br>Repetition rate 20 kHz<br>Optical efficiency 38.5% | [46] |
| 2015 | Huabei Photoelectric Technology Research Institute | $5 \times 5$ mm$^2$ end face | 2.05 μm Ho: YLF | Output wavelength 3.75 μm<br>Average power 26.9 W<br>Repetition rate 5 kHz<br>Optical efficiency 50% | [47] |
| 2016 | French-German Research Institute of Saint-Louis | $14 \times 12 \times 6$ mm$^3$ | 2.05 μm Ho: YLF | Output wavelength 4.6 μm<br>Average power 0.12 W<br>Repetition rate 1 Hz<br>Optical efficiency 78% | [48] |
| 2017 | Chinese Academy of Sciences | $6 \times 6 \times 15$ mm$^3$ | 2.09 μm Ho: YAG | Output wavelength 4.6 μm<br>Average power 95 mW<br>Repetition rate 5 Hz<br>Optical efficiency 75.7% | [49] |
| 2018 | Harbin Institute of Technology | 30 mm length | 2.05 μm Ho: GdVO$_4$ | Output wavelength 4.39 μm<br>Average power 2.05 W<br>Repetition rate 5 kHz<br>Optical efficiency 74.6% | [50] |
| 2019 | Harbin Institute of Technology | $6 \times 6 \times 20$ mm$^3$ | 2.09 μm Ho: YAG | Output wavelength 4.57 μm<br>Average power 103 W<br>Repetition rate 10 kHz<br>Optical efficiency 44.2% | [51] |
| 2019 | Changchun University of Science and Technology | $5 \times 5 \times 16$ mm$^3$ | 2.09 μm Ho: YAG | Output wavelength 4.5 μm<br>Average power 5.97 W<br>Repetition rate 6 kHz<br>Optical efficiency 44.1% | [52] |
| 2021 | French-German Research Institute of Saint-Louis | $6 \times 6 \times 20$ mm$^3$ | 2.09 μm Ho:LLF MOPA | Output wavelength 3–5 μm<br>Average power 38 W<br>Repetition rate 10 kHz<br>Optical efficiency 46.6% | [53] |
| 2021 | Shandong University | $6 \times 6 \times 25$ mm$^3$ | 2.1 μm Ho:YAG | Output wavelength 4.3 μm<br>Average power 10.62 W<br>Repetition rate 15 kHz<br>Optical efficiency 37.9% | [54] |

**Table 6.** Properties of mid-infrared nonlinear crystals mentioned above.

| Crystal | Crystal System | Point Group | Nonlinear Coefficient/pm·V$^{-1}$ | Transparency Range/μm | Thermal Conductivity/W·m$^{-1}$·K$^{-1}$ | Damage Threshold/GW·cm$^2$ |
|---------|----------------|-------------|-----------------------------------|-----------------------|------------------------------------------|----------------------------|
| LiNiO$_3$ | trigonal system | 3 m | $d_{22} = 2.1$ $d_{31} = 4.3$ $d_{33} = 27.2$ | 0.35–4.5 | 5.6 | 0.2 |
| PPLN | trigonal system | 3 m | $d_{33} = 27.2$ | 0.33–5.5 | 5 | 0.3 |
| MgO: PPLN | trigonal system | 3 m | $d_{13} = 14.8$ | 0.36–5 | 4.4 | 0.6 |
| KTP | orthorhombic system | 2 m | $d_{15} = 1.9$ $d_{24} = 3.64$ $d_{33} = 16.9$ | 0.35–4.5 | 0.4 | 1.5 |
| KTA | orthorhombic system | 2 m | $d_{15} = 4.2$ $d_{24} = 2.8$ $d_{33} = 16.2$ | 0.4–5 | 20 | 1.0 |
| AgGaSe$_2$ | tetragonal system | 42 m | $d_{36} = 39.5$ | 0.73–18 | 1 | 0.04 |
| AgGaS$_2$ | tetragonal system | 42 m | $d_{36} = 13.4$ | 0.53–13 | 1.5 | 0.04 |
| ZGP | tetragonal system | 42 m | $d_{eff} = 75$ | 0.74–12 | 35 | 30 |

## 3. Mid-Infrared Fe: ZnSe and Cr: ZnSe Solid-State Lasers

Taking transition metal doped II~VI chalcogenides crystallized group sulfide crystals as gain media is an important means to realize mid-infrared laser. The two typical laser materials are Fe: ZnSe and Cr: ZnSe crystals. Characteristics descriptions of Fe: ZnSe and Cr: ZnSe crystals are shown in Figure 6.

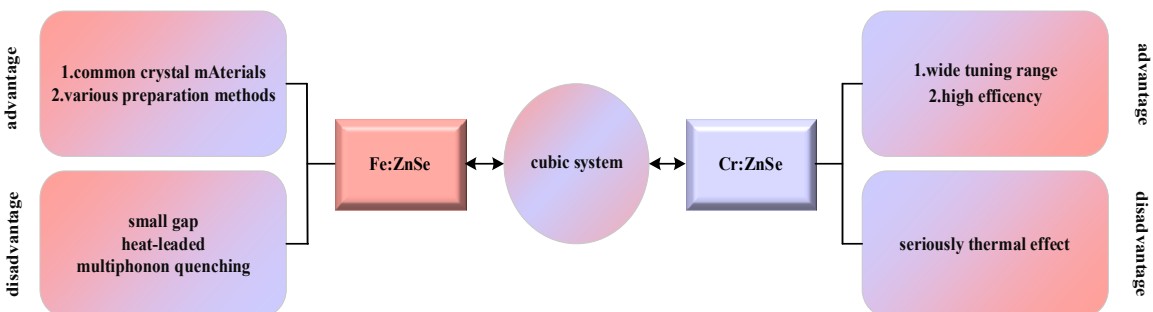

**Figure 6.** Characteristics descriptions of Fe: ZnSe and Cr: ZnSe crystals.

Fe: ZnSe is a four-energy level structure. When $Fe^{2+}$ is doped into ZnSe, $Zn^{2+}$ in the center of tetrahedron will be replaced. The ground state energy level $^5D$ of the outermost electron $^3d_6$ splits into duplex degenerate states $^5E$ and triple-degenerate states $^5T_2$ under the action of a crystal field [55]. Then the one-step orbital spin coupling splits the $^5T_2$ state into three energy bands and the second-order orbital spin coupling splits the $^5E$ state into five energy levels. The energy level diagram is shown as Figure 7.

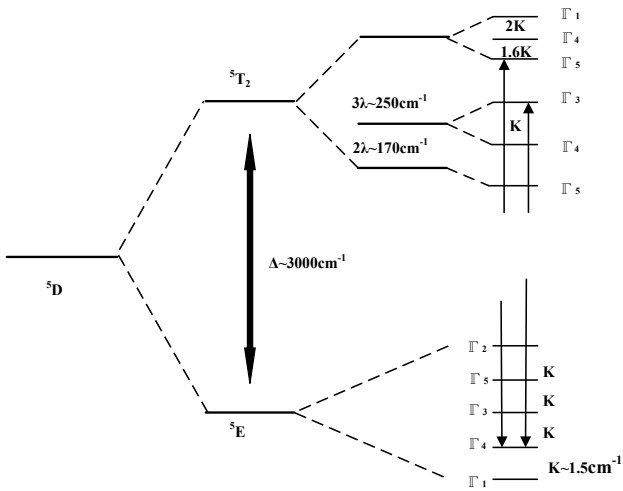

**Figure 7.** Diagram of Fe: ZnSe energy level.

Cr: ZnSe is a four-energy level structure. Under the action of a pump light, $Cr^{2+}$ in the ground state of $^5T_2$ transits to the vibrational levels of excited state $^5E$, and because there is no other energy level above the $^5E$ excited state level, therefore, there is almost no excited state absorption process for $Cr^{2+}$ [56]. The energy level diagram is shown as Figure 8.

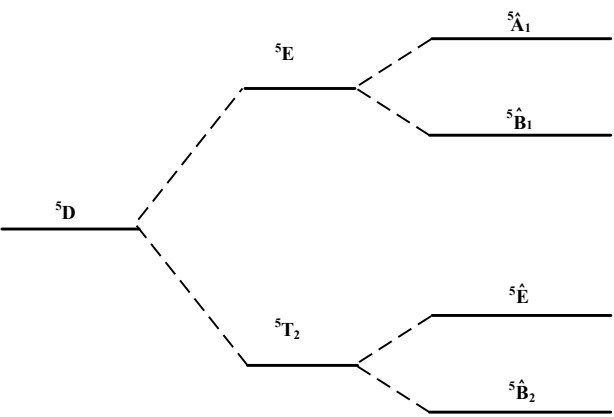

**Figure 8.** Diagram of Cr: ZnSe energy level.

The absorption peak of Fe: ZnSe crystal is near 3 μm at room temperature. Additionally, the emission peak is near 4.3 μm. Take note that the absorption characteristics of Fe: ZnSe crystal varies greatly with temperature, as shown in Figure 9. The absorption cross sections of Fe: ZnSe crystal are greatly at 14 K. Additionally, the absorption cross section will become lower while, at the same time, the absorption range will become wider at 300 K. From the emission spectrum of Fe: ZnSe, the material emission spectrum range is 3–5 μm [1].

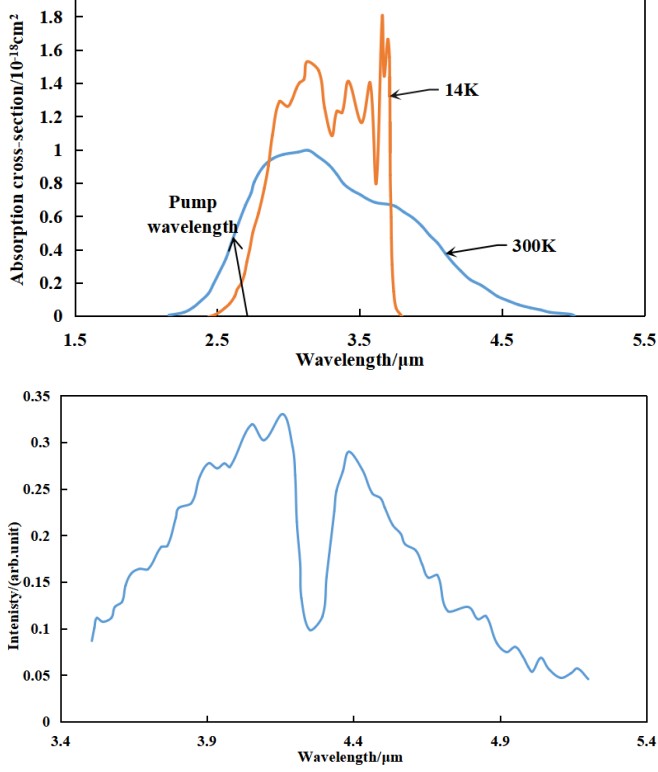

**Figure 9.** Absorption and emission spectrum of Fe: ZnSe crystal.

Cr: ZnSe has a relatively wide absorption band, at 1.5–2 μm; as shown in Figure 10, the absorption peak is around 1.75 μm. The emission spectroscopy is 2–3 μm, and the emission peak is about 2.45 μm [56]. It can be seen from Figure 10 that it is not a good choice to use the Cr: ZnSe crystal to achieve a laser output above 3 μm, because, although the crystal has emission at 3 μm, its gain is relatively low.

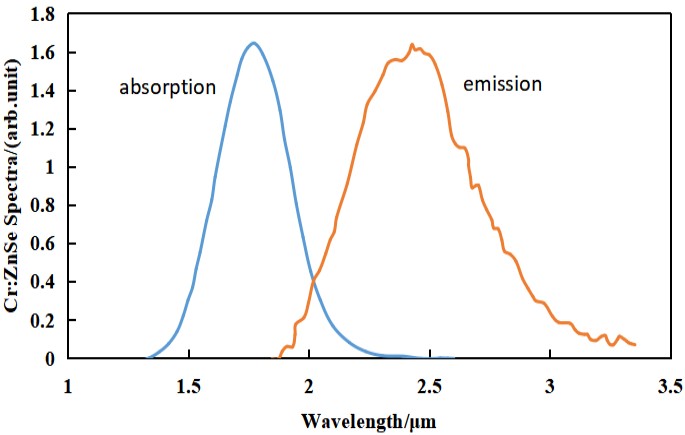

**Figure 10.** Absorption and emission spectrum of Cr: ZnSe crystal.

Spectroscopic and material properties of the Cr: ZnSe and Fe: ZnSe crystals are shown in Table 7.

**Table 7.** Parameters of Cr: ZnSe and Fe: ZnSe crystals.

| Crystal | Cr:ZnSe | Fe:ZnSe |
|---|---|---|
| Symmetry of crystal | Cubic system | Cubic system |
| Size (mm$^3$) | $40 \times 40 \times 50$ | $40 \times 40 \times 50$ |
| Launch range (μm) | 1.9–3.3 | 3.4–5.2 |
| Gain bandwidth (nm) | 500 | 500 |
| Peak absorption cross section ($\times 10^{-20}$ cm$^2$) | 87 | 97 |
| Peak absorption wavelength (μm) | 1.78 | 3 (300 K) |
| Peak emission cross section ($\times 10^{-20}$ cm$^2$) | 90 | 140 |
| Peak emission wavelength (μm) | 2.45 | 4.140 |
| Emission bandwidth (nm) | 0.9 | 1.1 |
| Fluorescence lifetime (300 k, μs) | 8 | 0.37 |

It can be seen from Table 7 that the absorption cross section and emission cross section of $Fe^{2+}$: ZnSe are larger than that of $Cr^{2+}$: ZnSe. While the Cr: ZnSe crystal exhibits excellent room temperature fluorescence properties, both of them have a wide tuning range and high quantum efficiency, which have attracted more and more attention in the field of mid-infrared wave band research. The research and development status of Cr: ZnSe and Fe: ZnSe lasers are shown in Table 8.

Compared with Cr: ZnSe laser, the single energy or the average power is higher for the Fe: ZnSe laser. However, for the Fe: ZnSe crystal, the temperature is the key factor affecting its fluorescence lifetime. High-power Fe: ZnSe laser can be realized at low temperatures. As temperature rises, the fluorescence lifetime of Fe: ZnSe crystal decreases, which makes it difficult to achieve a high-power, mid-infrared laser. Future research can focus on the external cooling method of the laser to ensure that it maintains good mid-infrared laser output performance at room temperature.

**Table 8.** Research and development status.

| Crystal | Year | Research Establishment | Crystal Parameter | Pump Source | Mid-Infrared Output Parameter | References |
|---|---|---|---|---|---|---|
| Fe:ZnSe | 2011 | University of Alabama at Birmingham | $8 \times 8 \times 3$ mm$^3$ $2 \times 10^{19}$ cm$^{-3}$ | 2.8 μm Er, Cr: YSGG | Temperature 300 k (0.38 μs) Output wavelength 4.3 μm Average power 0.3 mW Optical efficiency 16% Temperature 236 k (0.274 μs) Output wavelength 4.37 μm Average power 24.12 mW Optical efficiency 19% | [57] |
| | 2012 | Air Force Research Laboratory | $2 \times 6 \times 8$ mm$^3$ $9 \times 10^{18}$ cm$^{-3}$ | 2.94 μm Er: YAG | Temperature 300 k (0.37 μs) Output wavelength 4.14 μm Average power 840 mW Optical efficiency 39% | [58] |
| | 2013 | Russian Academy of Sciences | $8 \times 8 \times 8$ mm$^3$ $2.6 \times 10^{18}$ cm$^{-3}$ | 2.9 μm Er: YAG | Temperature 245 k (1.7 μs) Output wavelength 4.5 μm Average power 2.1 W Optical efficiency 23% | [59] |
| | 2015 | Heriot-Watt University | $1.82 \times 4.76 \times 6.94$ mm$^3$ $8.8 \times 10^{18}$ cm$^{-3}$ | 2.94 μm Er: YAG | Temperature 275 k (0.715 μs) Temperature 292 k (0.36 μs) Temperature 77 K (0.57 μs) Output wavelength 4.122 μm Average power 76 mW Optical efficiency 11% | [60] |
| | 2015 | University of Alabama | 2 mm thickness | 2.94 μm Er: YAG | Temperature 300 K (0.37 μs) Output wavelength 4.1 μm Average power 35 mW Optical efficiency 35% | [61] |
| | 2017 | All-Russian Research Institute of Experimental Physics | $120 \times 64 \times 4$ mm$^3$ $(7–9) \times 10^{18}$ cm$^{-3}$ | 2.6 μm HF | Temperature 300 k (0.36 μs) Output wavelength 4.3 μm Average power 20 W | [62] |
| | 2018 | Russian Academy of Sciences | $25 \times 25 \times 16.7$ mm$^3$ $1.1 \times 10^{18}$ cm$^{-3}$ | 2.94 μm Er: YAG | Temperature 80 k (60 μs) Temperature 220 k (8 μs) Temperature 250 k (3 μs) Temperature 300 k (0.37 μs) Output wavelength 4.3 μm Average power 7.5 W Optical efficiency 30% | [63] |
| | 2019 | Russian Academy of Sciences | 12 Diameter $\times$ 17 thickness mm$^3$ $1.8 \times 10^{18}$ cm$^{-3}$ | 2.94 μm Er: YAG | Temperature 5–18 °C (0.68–0.39 μs) Output wavelength 4.7 μm Average power 3.14 W Optical efficiency 17.5% | [64] |
| | 2019 | Harbin Institute of Technology | $4 \times 4 \times 10$ mm$^3$ $5 \times 10^{18}$ cm$^{-3}$ | 2.958 μm Ho, Pr: LLF | Temperature 77 k (0.57 μs) Output wavelength 3.957 μm Average power 0.0164 mW Optical efficiency 22.9% | [65] |
| | 2019 | Harbin Institute of Technology | $4 \times 10 \times 10$ mm$^3$ $5 \times 10^{18}$ cm$^{-3}$ | 2.93 μm Cr, Er: YAG | Temperature 77 k (0.57 μs) Output wavelength 4.037 μm Average power 197.6 mW Optical efficiency 13.7% Temperature 300 k (0.37 μs) Output wavelength 4.509 μm Average power 3.5 mW Optical efficiency 0.27% | [66] |
| | 2020 | Osaka University | 8 length $3.5 \times 10^{18}$ cm$^{-3}$ | 2.8 μm Er: ZBLAN | Temperature 77 k (57 μs) Output wavelength 4 μm Average power 880 mW Optical efficiency 44.2% | [67] |

**Table 8.** *Cont.*

| Crystal | Year | Research Establishment | Crystal Parameter | Pump Source | Mid-Infrared Output Parameter | References |
|---|---|---|---|---|---|---|
| | 2020 | Lomonosov Moscow State University | 8 length $3.5 \times 10^{18}$ cm$^{-3}$ | 2.8 µm Er: ZBLAN | Temperature 170 k Output wavelength 4.4 µm Average power 415 mW Optical efficiency 5.92% | [68] |
| | 2020 | Changchun Institute of Optics, Fine Mechanics and Physics | 28 mm diameter 4 mm thickness $2 \times 10^{18}$ cm$^{-3}$ | 2.6 µm HF | Temperature 300 k (0.37 µs) Output wavelength 3.1 µm Average power 21.7 W Optical efficiency 32.6% | [69] |
| | 2021 | University of Alabama at Birmingham | 2–3 mm length $1.5 \times 10^{19}$ cm$^{-3}$ | 2.94 µm Er: YAG | Temperature 120 k (57 µs) Temperature 300 k (0.37 µs) Output wavelength 4.1 µm Average power 180 mW Optical efficiency 25% | [70] |
| Cr: ZnSe | 2006 | Koç University | 2 mm thickness $5.7 \times 10^{18}$ cm$^{-3}$ | 1.57 µm KTP OPO | Temperature 300 k (5 µs) Output wavelength 3.1 µm Average power 145 mW Optical efficiency 8% | [71] |
| | 2007 | University of Alabama at Birmingham | $4 \times 8 \times 1$ mm$^3$ No mention | 1.55 nm Er$^{3+}$ fiber | Output wavelength 3 µm Average power 150 mW | [72] |
| | 2010 | Norwegian University of Science and Technology | 2.3 thickness mm $5 \times 10^{18}$ cm$^{-3}$ | 1.607 µm Er$^{3+}$ fiber | Output wavelength 3.3 µm Average power 600 mW | [73] |
| | 2021 | Tokyo University of Science | 5 length mm $8 \times 10^{-18}$ cm$^{-3}$ | 2.01 µm Tm:YAG | Output wavelength 3.2 µm Average power 49.8 mW Optical efficiency 22.5% | [74] |

## 4. Mid-Infrared Fiber Lasers

Optical fiber has many advantages in numerous fields. This paper mainly discusses the mid-infrared fiber laser with soft glass [fluoride (Er$^{3+}$, Ho$^{3+}$, Dy$^{3+}$), chalcogenide, telluride] as the gain medium. The description is shown in Figure 11.

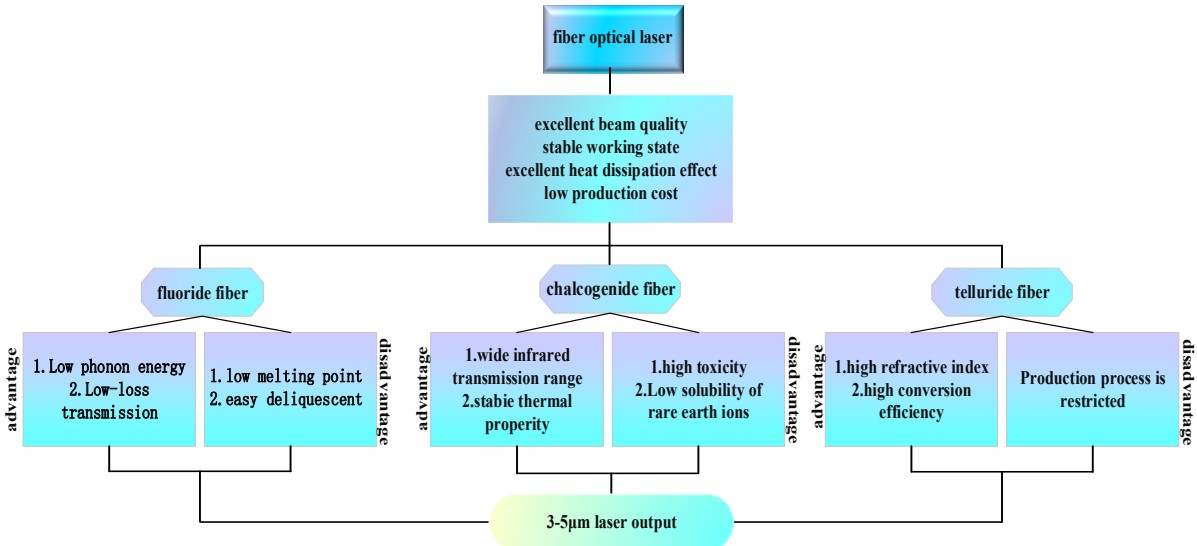

**Figure 11.** Description diagram of mid-infrared fiber lasers.

The most-used material for fluoride optical fiber is a multi-component fluoride glass called "ZBLAN"; the mid-infrared fiber laser operating at 3–5 μm band has a similar outer electron arrangement for gain ions. Energy level transitions between configurations produce abundant emission lines; the gain fiber mainly includes $Er^{3+}$, $Ho^{3+}$, $Dy^{3+}$, and its energy level diagram [75] is shown in Figure 12.

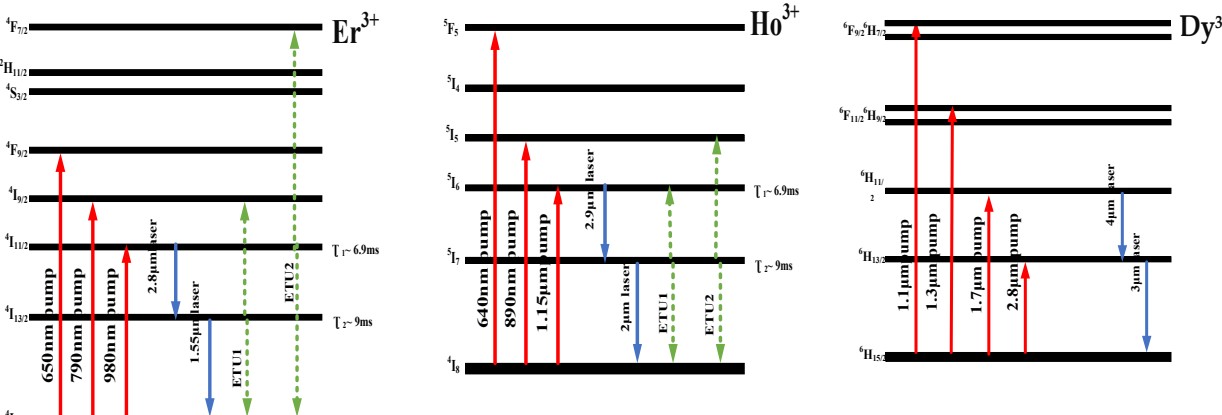

**Figure 12.** Energy level diagram.

The chalcogenide glass has excellent mid-infrared transmission, thermal and mechanical properties. Compared with fluoride glass fiber, its phonon energy is lower, which makes up for the defect that ZBLAN is hindered to work at wavelengths exceeding 4 μm due to the reduction of high-energy states caused by multi-phonon transitions. In the context of the chalcogenide glass fiber lasers, the ions that have received the most attention are praseodymium and terbium. The energy level diagram [76] is shown in Figure 13.

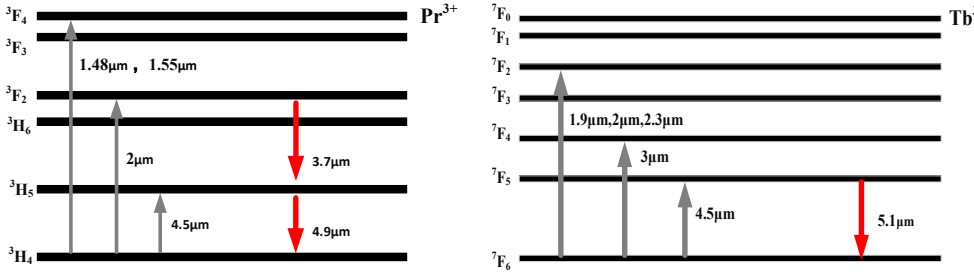

**Figure 13.** Energy level diagram.

For the glass fibers of fluoride, chalcogenide and tellurite, their physical and chemical properties are different, as shown in Table 9.

**Table 9.** Physicochemical properties of various soft glass fibers.

| Properties | Fluoride | Chalcogenide | Tellurite |
|:---:|:---:|:---:|:---:|
| The lowest loss (dB/m) | $0.45 \times 10^{-3}$ | 0.023 | 0.02 |
| Max. phonon energy (cm$^{-1}$) | 560 | 300–450 | 700 |
| Transparency (μm) | 0.4–6 | 1–16 | 0.5–5 |
| Nonlinear refractive index ($\times 10^{-20}$ m$^2$/W) | 2–3 | 300–500 | 59 |
| Melting point (°C) | 265 | 250 | 500 |
| Durability | poor | good | good |
| Toxicity | high | high | safe |

Compared with chalcogenides, the fluoride glass has lower loss but higher phonon energy, and its transparency range is far inferior to chalcogenide's. However, compared with tellurite glass, the fluoride glass and chalcogenide glass are more toxic. Three kinds of glass optical fibers are the best choice for mid-infrared transmission. Their low optical loss and high-power damage threshold make many applications possible.

The fiber lasers with different gain media have unique advantages and characteristics. The developments are shown in Table 10.

**Table 10.** Research and development status.

| Medium | Fiber Matrix | Year | Research Establishment | Crystal Parameter | Pump Source | Mid-Infrared Output Parameter | References |
|---|---|---|---|---|---|---|---|
| Fluoride | Er: ZBLAN | 2014 | The University of Adelaide | 18 mm length 1 %mol | 1.973 μm fiber laser | Output wavelength 3.5 μm Average power 260 mW Optical efficiency 16% | [77] |
| | | 2016 | Chinese Academy of Sciences | 0.9 mm length 6% mol | 0.975 μm LD pump beam | Output wavelength 3 μm Average power 1.01 W Repetition rate 146.3 kHz Optical efficiency 17.8% | [78] |
| | | 2018 | Shanghai Jiao Tong University | 2.8 mm length 1% mol | 1.973 μm $Tm^{3+}$ fiber | Output wavelength 3.489 μm Average power 40 mW Repetition rate 28.91 MHz Optical efficiency 18% | [79] |
| | | 2019 | Université Laval | 2.5 mm length 7% mol | 976 + 1976 nm LD pump beam | Output wavelength 3.42 μm Average power 3.4 W Optical efficiency 38.6% | [80] |
| | | 2020 | University of Electronic Science and Technology of China | 3.2 mm length 1.5% mol | 976 + 1981 nm LD pump beam | Output wavelength 3.45 μm Average power 264.5 mW Optical efficiency 7.18% | [81] |
| | | 2021 | Shenzhen University | 1.8 mm length 1% mol | 976 + 1973 nm LD pump beam | Output wavelength 3.46 μm Average power 63 mW Repetition rate 58.71 MHz Optical efficiency 15.6% | [82] |
| | Ho: ZBLAN | 2011 | University of Sydney | 10 mm length 1.2% mol | 1.15 μm LD pump beam | Output wavelength 3.002 μm Average power 77 mW Optical efficiency 12.4% | [83] |
| | | 2012 | University of Electronic Science and Technology of China | 12 mm length 1.2% mol | 1.15 μm LD pump beam | Output wavelength 3.005 μm Average power 175 mW Repetition rate 75 kHz | [84] |
| | | 2013 | University of Arizona | 2.5 mm length 3% mol | 1.15 μm Roman laser | Output wavelength 3 μm Average power 100 mW Repetition rate 100 kHz Optical efficiency 12.3% | [85] |
| | Ho:InF$_3$ | 2018 | Université Laval | 2.3 mm length 10% mol | 888 nm LD pump beam | Output wavelength 3.92 μm Average power 197 mW Optical efficiency 9.77% | [86] |
| | | 2021 | University of Electronic Science and Technology of China | 0.23 mm length 10% mol | 888 + 974 nm LD pump beam | Output wavelength 3.92 μm Average power 1.3 W Optical efficiency 21.6% | [87] |
| | Dy: ZBLAN | 2016 | Macquarie University | 0.92 mm length 2000 ppm | 2.8 μm Er: ZBLAN | Output wavelength 3.04 μm Average power 80 mW Optical efficiency 51% | [88] |
| | | 2016 | Macquarie University | 0.14 mm length 2000 ppm | 2.8 μm Er: ZBLAN | Output wavelength 3.26 μm Average power 120 mW Optical efficiency 37% | [88] |
| | | 2018 | Macquarie University | 0.6 mm length 2000 ppm | 1.7 μm Raman laser | Output wavelength 3.4 μm Average power 170 mW Optical efficiency 21% | [89] |
| | | 2019 | Université Laval | 2.2 mm length 2000 ppm | 2.83 μm Er: ZBLAN | Output wavelength 3.24 μm Average power 10.1 W Optical efficiency 58% | [90] |

**Table 10.** *Cont.*

| Medium | Fiber Matrix | Year | Research Establishment | Crystal Parameter | Pump Source | Mid-Infrared Output Parameter | References |
|---|---|---|---|---|---|---|---|
| | | 2020 | Université Laval | 1.75 mm length 2000 ppm | 2.825 μm Er: ZBLAN | Output wavelength 3.24 μm Average power 1.43 W Repetition rate 120 kHz Optical efficiency 22% | [91] |
| | Dy:InF$_3$ | 2021 | University of Electronic Science and Technology of China | 1.25 mm length 0.1% mol | 1.1 μm Yb$^{3+}$:fiber laser | Output wavelength 4.3 μm Average power 107 mW Optical efficiency 3.75% | [92] |
| | As$_2$S$_3$ | 2014 | Université Laval | 2.8 mm length 98% reflectivity | 3.005 μm Er: ZBLAN | Output wavelength 3.77 μm Average power 112 mW Optical efficiency 8.3% | [93] |
| | As$_2$Se$_3$ | 2019 | Ningbo University | 1.05–1.23 mm length 97.8–98% reflectivity | 3.92 μm Ho$^{3+}$:InF$_3$ | Output wavelength 4.327 μm Average power 0.269 mW Optical efficiency 17.9% | [94] |
| | Dy$^{3+}$: GGSS | 2019 | Chinese Academy of Sciences | 120 mm length Dy$^{3+}$:0.3 wt% 125:60:11 /125:66:11.5 core/cladding | 1.7 μm Tm$^{3+}$:fiber laser | Output wavelength 4.21 μm Impurity absorption peaks 2.4 dB/m $\sigma_e \times \tau_{mea}$ 2.62 × 10$^{-23}$ cm$^2$ Lifetime 4.61 ms | [95] |
| Chalcogenide | Tb$^{3+}$: GGS | 2020 | Russian Academy of Sciences | 12 mm diameter 56 mm length 2 × 10$^{19}$ cm$^{-3}$ | 2.93 μm Er:YAG laser | Output wavelength 4.9–5.5 μm $\sigma_e(\lambda)$ = 5 × 10$^{-21}$ cm$^2$ Average power 25 mW Lifetime 10 ms | [96] |
| | Ce$^{3+}$: GSGS | 2021 | Russian Academy of Sciences | 12 mm diameter 24 mm length 3 × 10$^{19}$ cm$^{-3}$ | 4.08 μm Fe:ZnSe laser | Output wavelength 5 μm Energy output 0.5 mJ Impurity absorption 6 × 10$^{-3}$ cm$^{-1}$ Lifetime 3.7 ms | [97] |
| | Ce$^{3+}$: GSGS | 2021 | University of Duisburg-Essen | 12 mm diameter 24 mm length 3 × 10$^{19}$ cm$^{-3}$ | 4.1 μm Fe:ZnSe laser | Output wavelength 5.2 μm Energy output 35 mJ Optical efficiency 21% | [98] |
| | Ce$^{3+}$: GAGS | 2021 | University of Nottingham | 9 μm diameter 64 mm length 500 ppmw | 4.15 μm quantum cascade laser | Output wavelength 4.63 μm Impurity absorption peaks 2.16 dB/m$^{-1}$ Lifetime 3.6 ms | [99] |
| | Pr$^{3+}$: GGS | 2021 | Institute of Chemistry of High-Purity Substances | 12 mm diameter 5 mm length 1 × 10$^{20}$ cm$^{-3}$ | 1.54 μm Er:glass laser | Output wavelength 5.5 μm Average power 20 mW Lifetime 3 ms | [100] |
| | | 2015 | University of Arizona | 1 mm length 10–20% reflectivity | 2.8 μm Er: ZBLAN | Output wavelength 3.16 μm Average power 7.42 W Optical efficiency 7.55% | [94] |
| | | 2017 | Hefei University of Technology | 0.3 mm length 90% reflectivity | 2 μm fiber laser | Output wavelength 3.64 μm Average power 45.2 W Optical efficiency 45.2% | [101] |
| Tellurite | TBZN | 2018 | National University of Defense Technology | 5.5 mm length 45% reflectivity | 2 μm Tm$^{3+}$:fiber laser | Output wavelength 3.61 μm Average power 16 W Optical efficiency 45.2% | [102] |
| | | 2021 | University of Electronic Science and Technology of China | 0.2 mm length 69% reflectivity | 1.96 μm Tm$^{3+}$:fiber laser | Output wavelength 5 μm Average power 52.44 mW Optical efficiency 19% | [103] |

From the current research progress, the soft glass fiber (fluoride, chalcogenide and telluride) has low loss in the mid-infrared band. The manufacturing process is relatively mature. Therefore, achieving mid-infrared laser with fiber has been extensively studied by scholars. Among the soft glass fibers, the manufacturing process of ZBLAN fiber

is relatively mature. However, the realization of mid-infrared laser output with high conversion efficiency and the output energy still needs further development; due to the limited manufacturing process of $InF_3$ and the telluride, there are still difficulties in general commercial use; chalcogenide glass has excellent transmission performance in the mid-infrared band due to its low material dispersion, so it has an indispensable application value at 3–5 μm. For the future, it is necessary to optimize the gain fiber, to increase the pump power and to achieve a higher power mid-infrared laser output.

**5. Conclusions**

In the past 20 years, based on the progress of new laser materials, optical technology and the traction of application requirements in many fields, the research of mid-infrared laser has made many breakthroughs and rapid progress. In order to improve the performance of mid-infrared lasers, it is urgent to study and improve the physical and chemical properties of the gain medium for achieving mid-infrared laser output and develop technologies to improve the performance of mid-infrared lasers. In general, the paper briefly introduces the development of mid-infrared optical parametric oscillators, direct-pumped mid-infrared solid-state lasers and direct lasing mid-infrared fiber lasers. Looking forward to the future, the main development trends mainly include: (1) output power increases; in the future, we can continue to improve mid-infrared laser technology and soft glass pretreatment and find new gain media to continuously increase the output power of 3–5 μm mid-infrared laser and (2) lift the conversion efficiency furthermore; with the low-loss beam-coupling technology development and the successful development of lower loss optical fiber, based on the improvement of passive $InF_3$ fiber and chalcogenide purification technology, it can be expected that there is still room for improvement in conversion efficiency.

We can expect that, in the near future, with the continuous improvement of various technologies, the high-power, large-energy mid-infrared laser of 3–5 μm will move from experimental research to practical applications which will play a unique role in scientific research and production.

**Author Contributions:** Writing—original draft preparation and writing—review and editing, T.R.; methodology, C.W.; funding acquisition, Y.Y.; supervision, F.C., T.D. and Q.P. All authors have read and agreed to the published version of the manuscript.

**Funding:** Science and Technology Department of Jilin Province in China (Grant No. 202002041JC).

**Acknowledgments:** We thank the Key Laboratory of Jilin Province Solid-State Laser Technology and Application for the use of the equipment.

**Conflicts of Interest:** The authors declare no conflict of interest.

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
