# Peer review of "Development Progress of 3–5 μm Mid-Infrared Lasers: OPO, Solid-State and Fiber Laser"

_applsci, doi:10.3390/app112311451_

Round 1

Reviewer 1 Report

This is a very timely contribution. A large progress has been made over the last decade in this area and a review paper is very helpful. In general the paper is well written but some improvements still could be made. I suggest the following changes:

  1. The title should contain a statement saying that the wavelength range is limited to 3-5 mi. MIR wavelengths contain a larger range of wavelengths cf. for instance: 
  • https://doi.org/10.3390/fib6020025
  • https://doi.org/10.1364/OL.430891

2.The authors could consider mentioning chalcogenide glass as a potential material for realization of lasing. Papers: 

  • https://doi.org/10.3390/fib6020025
  • https://doi.org/10.1364/OL.430891
  • https://doi.org/10.1016/j.jnoncrysol.2020.120592
  • https://doi.org/10.1007/s00340-020-07473-w

show the potential of this technology. Though I admit for wavelengths mostly exceeding 5 mi. However, for wavelengths between 4-5 mi this material is very promising thanks to low phonon energy. Most likely some impurities prevent lasing in this wavelength range and lasing only beyond 5 mi has been observed very recently. However, theory shows that lasing between 4 and 5 mi should be possible. 

3. I suggest another read to remove minor typos and to improve style in some parts of the manuscript. For instance line 197. It should be fluoride glass I guess? Then sulfide glass I guess? Also line 181 'lots of' replace with 'many' to make the text sound more formal.

4. It would be also helpful if the authors were more descriptive when discussing the results presented in tables and figures. These tables and figures are great and mostly self descripting but still more text would be helpful.

Author Response

To the reviewer 1

In reply to your suggestion, I made rapid revisions to my paper, the specific modifications are as follows:

  1. The title has been modified according to your suggestion, the new title is: Development progress of 3-5 μm mid-infrared lasers: OPO, solid-state, and fiber laser.
  2. According to your suggestion, a new part of chalcogenide glass fiber has been added to the paper, and the references you suggested have also been quoted in the paper.
  3. The article has been read and revised again, include the fluoride glass and so on.
  4. According to your suggestions, the figures and tables in the paper have been described in detail.

Thanks again for your effort on my paper.

Reviewer 2 Report

In the paper entitled “Review of mid-Infrared OPO, solid-state, and fiber lasers”, Tingwei Ren et al propose a review of different technologies for laser emission in the mid-IR.

This region of the optical spectrum is interesting in many applications but remains inaccessible with conventional simple lasers and a review of the different technologies is therefore of high interest.

The paper is divided in different sections describing each technology.

Though interesting explanations are given for each type of technology, I would say that in a general way the numerous references are not sufficiently discussed. It would help the reader to give more information and put the focus on the most interesting/innovative works.

I would like also to point out some sentences in the paper that should be changed:

  • In the abstract, line 13, “3-5μm mid-infrared lasers have the advantages of high efficiency, small size and light weight”.

This is really strange because mid ir lasers are much less common than nir and visible lasers which generally exhibit small footprint, high power and moderate cost

  • In table 1, it is written that OPO exhibits simple structure and complex systems which is a nonsense.

  • Line 64, both acronyms MgO:PPLN and PPLN represents “periodically poled lithium niobate”   

  • There are some mistakes in different tables. For instance, in table 2 the average power is 7 mJ while this value is only related to pulse energy. Average power has to be defined in W/mW and pulse energy in µJ/nJ.

Author Response

To Reviewer 2

In reply to your suggestion, I made rapid revisions to my paper, the specific modifications are as follows:

  1. According to your suggestion, the question you raised in the abstract has been revised. The expression that “3-5μm mid-infrared band is a good window for atmospheric transmission. It has the advantages of high contrast and strong penetration under high humidity conditions.” has been inserted in the paper.
  2. The description of OPO in table 1 has been changed to“The advantageis high energy, efficiency and excellent spectral characteristics and disadvantage is system stability and beam quality should be improved.”
  3. The expression about “MgO: PPLN” has been changed to “MgO-doped: PPLN” in the paper.
  4. The mistakes in tables have been modified carefully and the units have been revisedaccurately. 

  Thanks again for your effort on my paper.
